

# Comparison of the impacts of urban development and climate change in exposing European cities to pluvial flooding

Per Skougaard Kaspersen[1], Nanna Høegh Ravn[2], Karsten Arnbjerg-Nielsen[3], Henrik Madsen[4], Martin Drews[1].

[1]Department of Management Engineering, Technical University of Denmark, Kgs. Lyngby, 2800, Denmark
[2]LNH Water, Tikoeb, 3080, Denmark
[3]Department of Environmental Engineering, Technical University of Denmark, Kgs. Lyngby, 2800, Denmark
[4]DHI, Hoersholm, 2970, Denmark

*Correspondence to*: Per Skougaard Kaspersen (pskk@dtu.dk)

**Abstract.** The economic and human consequences of extreme precipitation and the related flooding of urban areas have increased rapidly over the past decades. Some of the key factors that affect the risks to urban areas include climate change, the densification of assets within cities and the general expansion of urban areas. In this paper, we examine and compare quantitatively the impact of climate change and recent urban development patterns on the exposure of four European cities to

pluvial flooding. In particular, we investigate the degree to which pluvial floods of varying severity and in different geographical locations are influenced to the same extent by changes in urban land cover and climate change. We have selected the European cities of Odense, Vienna, Strasbourg and Nice for analyses to represent, different climatic conditions, trends in urban development and topographical characteristics. We develop and apply a combined remote-sensing and flood-modelling approach to simulate the extent of pluvial flooding for a range of extreme precipitation events for historical (1984)

and present-day (2014) urban land cover and for two climate-change scenarios (RCP 4.5 and RCP 8.5). Changes in urban land cover are estimated using Landsat satellite imagery for the period 1984-2014. We combine the remote-sensing analyses with regionally downscaled estimates of precipitation extremes of current and expected future climate to enable 2D overland flow simulations and flood-hazard assessments. The individual and combined impacts of urban development and climate change are quantified by examining the variations in flooding between the different simulations along with the

corresponding uncertainties. For all four cities, we find an increase in flood exposure corresponding to an observed absolute growth in impervious surfaces of 7-12 % during the past thirty years of urban development. Similarly, we find that climate change increases exposure to pluvial flooding under both the RCP 4.5 and RCP 8.5 scenarios. The relative importance of urban development and climate change on flood exposure varies considerably between the cities. For Odense, the impact of urban development is comparable to that of climate change under an RCP 8.5 scenario (2081-2100), while for Vienna and

Strasbourg it is comparable to the impacts of an RCP 4.5 scenario. For Nice, climate change dominates urban development as the primary driver of changes in exposure to flooding. The variation between geographical locations is caused by



differences in soil infiltration properties, historical trends in urban development and the projected regional impacts of climate change on extreme precipitation.

## 1 Introduction

In recent years it has been widely demonstrated that cities globally are becoming increasingly exposed to the occurrence and impacts of pluvial flooding (Barredo, 2007; Field et al., 2012). It is evident that the observed changes in risk have been caused by a combination of different factors. These include ongoing climate change, leading to increases in the frequency and intensity of extreme precipitation events, general population growth and high rates of urbanisation during the late 20[th] and early 21[th] centuries. Thus, the extent of urban land cover has dramatically increased while at the same time rapidly increasing concentrations of assets and economic activities in cities worldwide (Angel et al., 2011; Field et al., 2012). Current trends in urban development and further urban densification, including soil-sealing, are projected to continue in all regions of the world (Angel et al., 2011; United Nations et al., 2014). As a result, urban areas are expected to become even more exposed and vulnerable to flooding in the future.

A key feature of most cities is the high proportion of impervious surfaces (IS), in the form of roads, buildings, parking lots and other paved areas. For this reason IS are often used as an indicator of urbanisation (Weng, 2012). Changes in the quantity and location of IS have important implications for the hydrological response of a catchment. Replacing natural land cover with sealed surfaces generally reduces infiltration capacity, surface storage capacity and evapotranspiration (Parkinson and Mark, 2005; Butler and Davies, 2011; Hall et al., 2014). Moreover, it leads to a loss of natural water retention and consequently increased run-off volumes, discharge rates, flood peaks and flood frequency (Butler and Davies, 2011). Knowing the exact quantity and location of IS is therefore important in estimating the spatially distributed run-off volumes during high-intensity rainfall.

Satellite imagery and remote-sensing techniques offer a complete spatial and temporal coverage of urban land cover changes during the past thirty to forty years, and may be used to quantify changes in IS. Medium resolution imagery, including Landsat, for example, has been found to provide accurate estimates of the quantity and distribution of IS with absolute mean errors < 10%, which is an acceptable level of accuracy for many applications (Chormanski et al., 2008; Verbeiren et al., 2013; Dams et al., 2013; Kaspersen et al., 2015). While high-resolution and hyperspectral imagery may be ideally suited for addressing the large heterogeneity of urban environments, the low temporal coverage (limited availability of historical archives), small scene sizes and high acquisition costs of these data-sets often constitute a barrier to their use in mapping the urban development of major cities over decadal time scales or longer (Weng, 2012; Verbeiren et al., 2013). Conversely, the application of the freely available medium-resolution Landsat imagery constitutes a cost- and resource-efficient alternative



for hydrological modelling purposes compared to traditional and high-resolution feature- or pixel-based techniques (Verbeiren et al., 2013).

The influence of changes in IS on urban hydrology has been investigated by quite a few authors (e.g. Weng, 2001;
Chormanski et al., 2008; Poelmans et al., 2010; Dams et al., 2013; Verbeiren et al., 2013; Urich and Rauch, 2014; Skougaard Kaspersen et al., 2015). Most of these studies adopt a modelling approach, where estimates of IS at two or more points in time are combined with spatially distributed hydrological models in order to examine the influence of changes in urban land cover on water balance parameters. Results generally confirm that the increase in urbanisation strongly affects peak discharges, run-off volumes and hydrological response times during extreme precipitation (Semadeni-Davies et al., 2008).
The impact of changes in imperviousness are found to be more pronounced for less extreme than very extreme (and probabilistically less frequent) events (Hollis, 1975). One reason for this difference is that pervious surfaces are more likely to become saturated during high-intensity rainfall events and thus begin to behave as impervious areas after the onset of the rain. The saturation time is longer during less extreme events. That said, limited research has been conducted examining and comparing the degree of variability across geographical locations on the relative impacts of soil-sealing (i.e. urban
development) and climate change in exposing entire urban areas to pluvial flooding.

In this paper, we examine the impacts of changes in urban land cover and climate change in exposing urban areas to pluvial flooding in four European cities. Our aim is to separate the importance of these drivers of changes to flood risks and compare them. To do so we quantify the impacts of the past thirty years of urban development in exposing urban areas to pluvial
flooding and compare them with expected climate change impacts on the four cities of Nice, Strasbourg, Vienna and Odense. These cities were carefully selected for analyses as they occupy different geographical locations across Europe and thus represent different climatic conditions, (expected) dissimilar historical trends in urbanisation and varying soil characteristics and topographies (flat vs. hilly), all of which are important for infiltration processes during extreme rain events. We expect that the impact of urban development is more pronounced for cities characterized by coarse soil textures and limited
topography, since soil infiltration rates are higher here, causing soil-sealing to have a greater impact on the urban hydrological response to extreme precipitation in such areas. In addition, due to longer soil saturation times during precipitation with shorter return periods (RPs), the influence of changes in imperviousness is likely to be more prominent for less intense precipitation, whereas they affect the hydrological response to the most extreme events to a lesser degree. By applying a systematic methodology, we estimate the relative importance of some of the main factors that expose urban areas
to pluvial flooding. Increased knowledge of these phenomena in different locations will enable local and national decision-makers to prioritize efficiently between different adaptation measures and urban development strategies when climate-proofing cities in the future.



## 2 Methods and data

### 2.1 Framework

A combined remote-sensing and flood-modelling approach is adopted to simulate the occurrence and extent of flooding following a range of design precipitation events under current and expected future climate conditions. To include the influence of spatial variations in urban land cover (changes in *IS*), simulations are performed for two different urban configurations corresponding to historical (1984) and present-day (2013-2015) observations of the urban surface.

The data-processing and evaluation procedures being carried out are divided into three separate types of analysis: (a) urban development analysis, (b) flood modelling and (c) quantification of the influence of urban development and climate change on exposure to flooding (Figure 1). Initially, we analyse Landsat TM (Landsat 5) (1984) and Landsat OLI (Landsat 8) (2013-2015) imagery to quantify IS fractions at a pixel level of 30m x 30m for historical and current urban land-cover conditions. Secondly, the outputs of the remote-sensing analyses are combined with soil infiltration data and regionally downscaled estimates of current and expected future precipitation extremes to enable 2D overland flow simulations and flood hazard assessments within a flood-modelling framework. The importance of two different assumptions about the temporal development of the drainage system were tested, that is, a "stationary" and an "evolutionary" scenario. In the latter we assume that the design of the drainage systems is updated to match the degree of urban development in the period 1984-2014 (2013-2015), as well as the projected intensity of extreme precipitation towards the end of this century (2081-2100) for RCP 4.5 and RCP 8.5 (Meinshausen et al., 2011), respectively. This is based on the notion that urban drainage systems are often developed in parallel to (planned) urban development and, in some cases, based on the future projections of precipitation from climate models (Chocat et al., 2007; Arnbjerg-Nielsen, 2011; Gregersen et al., 2014). This allows us to examine quantitatively the efficiency of expanding the urban drainage system, which is currently the most common large-scale adaptation measure in protecting urban areas from pluvial flooding. Flood hazard maps, representing the extent and depth of flooding, are generated for various combinations of urban land cover, intensities of extreme precipitation (i.e. different return periods) and climate scenarios. Finally, the relative influence of urban development and climate change for exposure to flooding is evaluated through a comparison of multiple flood hazard maps. The impact of recent urban development is isolated by simulating the occurrence of identical design precipitation events for both historical and current levels of urban *IS* fractions. Conversely, design precipitation intensities are varied, to reflect both current and expected future precipitation intensities, and imperviousness is kept constant while evaluating the expected impacts of different climate change scenarios. A total of 84 combinations of input parameters with regard to degree of imperviousness, climate scenario, climate model projection, precipitation return period, soil water infiltration and drainage system development were simulated for each city. Twelve of these were performed for historical (1984) levels of imperviousness, while an additional 72 were conducted for the present-day (2013-2015) cities. A pairwise cross-comparison of multiple flood-hazard maps is carried out to quantify the relative importance of changes in land cover as compared to climate change for the overall exposure to pluvial flooding. All





climate-change impact scenarios are simulated with *IS* corresponding to the present-day situation, for example, reflecting the increased hazards without implementing suitable adaptive measures as part of future urban development.

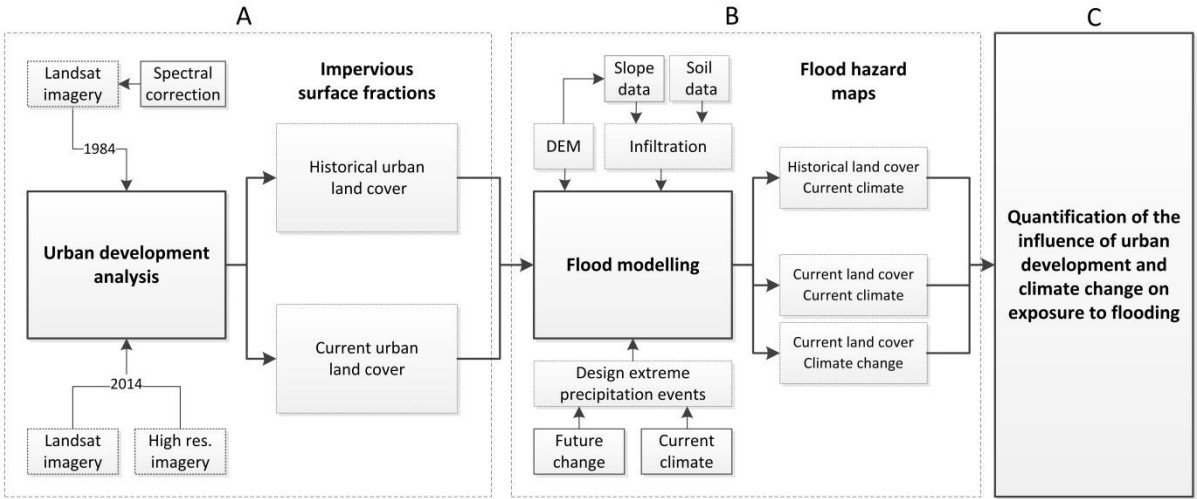

**Figure 1: Schematic of the methodology for quantifying the influence of changes to urban land cover and climate change on exposure to pluvial flooding.**

## 2.2 Urban development analysis

Since urban land-use is generally characterized by a large degree of heterogeneity (roads, buildings, vegetation, water and bare soil within small distances), it is often highly problematic to categorize and map urban structure and development at the desired scale accurately (Dams et al., 2013). For the purpose of this analysis, we define an "inner urban zone" for each city based on the land cover/land use classification in the CORINE 2012 Land Cover (CLC) Inventory (CORINE, 2012). The categories that are considered urban in this context are: *urban fabric*, *industrial and commercial units*, *port areas*, *airports*, *dump sites*, *green urban areas* and *sport and leisure facilities*. In the following, urban development and changes in flood exposure are considered only within this zone.

A number of techniques have been developed to estimate the quantity and location of IS as a proxy for urban land cover, including traditional field surveys with GPS, manual digitizing from hard-copy maps and, more recently, pixel- or feature based-methods using remotely sensed imagery (Weng, 2012). As most urban development occurs at decadal timescales, temporal coverage is a key parameter when monitoring changes to urban land cover. While state-of-the-art high-resolution satellite imagery (<5m) only dates back to the late 1990s and early 2000s, medium-resolution (MR) data (e.g. Landsat imagery) offer complete spatial and temporal coverage of global changes to urban land cover during the past thirty to forty years, from 1984 onwards, at a spatial resolution of 30m.




In this study, the quantification of spatio-temporal changes in imperviousness is based on an near-linear relationship between vegetation cover and IS fractions (Figure 2ab), which exists in many urban environments (Bauer et al., 2002; Lu et al., 2014; Kaspersen et al., 2015). Linear regression models (**Error! Reference source not found.**) developed by Kaspersen et al. (2015) relating the Soil Adjusted Vegetation Index (SAVI) to levels of imperviousness were applied to estimate IS fractions for the four cities both historically and at the present day.

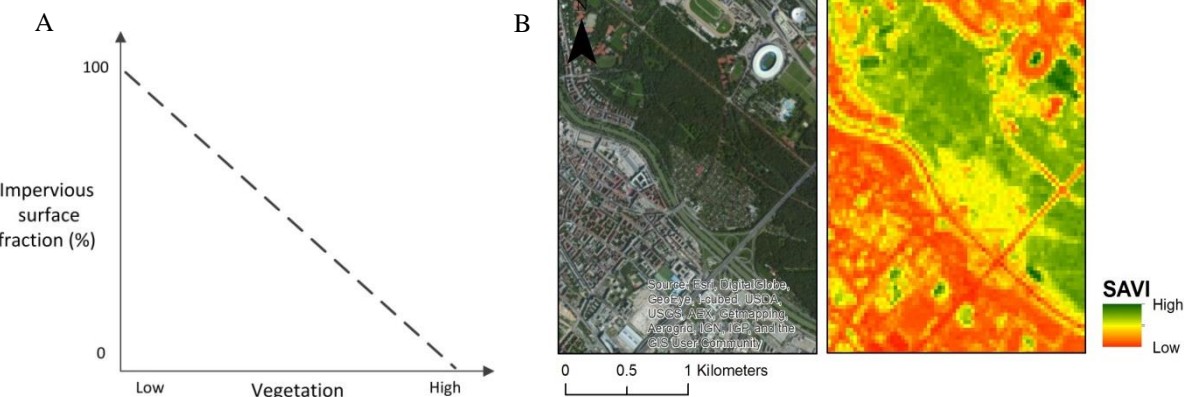

**Figure 2: (a) Conceptual relationship between impervious surface fractions and vegetation cover/vegetation indices in urban environments (adapted from Bauer et al., 2008); (b) example of high resolution image, used to measure impervious surface fractions and Soil Adjusted Vegetation Index (SAVI) calculated from Landsat OLI for a central part of Vienna.**

Multiple Landsat images are compiled into Maximum Value Composites (MVCs) to reduce the influence of inter-annual and intra-annual variations in the timing of maximum vegetation cover (**Error! Reference source not found.** in Supplementary Materials). Based on these composites, the impervious surface (IS) fractions for individual grid cells are calculated as:

$$IS = a * mvcSAVI + b \qquad\qquad (1)$$

where a and b are regression parameters to be estimated individually for each city.

Initial IS fractions were corrected for pixel values of <0% and >100% to restrict the output from the linear regression models that exceed the range of 0-100%. Impervious surface fractions are estimated at a spatial resolution of 30m, which is the same resolution as the short-wave (visual and near-infrared) bands in the Landsat TM and OLI sensors. These are later resampled to a spatial resolution of 25m, using a nearest neighbour assignment approach, to match the resolution of the Digital Elevation Model (DEM) used for overland flow calculations. In order for the reflectance data, and thus the subsequent SAVI images and IS fractions, to be comparable between the Landsat TM (historical images) and OLI (present-day) sensors, the difference in the spectral response function between the two sensors is corrected for by applying conversion factors to each of the individual bands of the TM sensor (or the OLI sensor depending on which the regression models are developed from)




(Flood, 2014). Further information on the different analytical steps involved in the processing of Landsat data may be found in Kaspersen et al. (2015).

## 2.3 Run-off and infiltration

Several empirical and theoretical methods have been developed to describe infiltration characteristics during precipitation of varying intensities, these being the key engines controlling overland run-off generation in hydrological models (Tomicic, 2015). Some of the most widely applied methods include Horton's equation, the Green-Ampt method and the Soil Conservation Service (SCS) method (Horton, 1933; Green and Ampt, 1911; USDA, 1986). All three methods assume that the infiltration rate decreases with time, but they use different equations (and input data) to determine the initial and final losses and the slope of the curve describing the infiltration rate. Even simpler techniques, where infiltration is defined as a constant or proportional loss, may be sufficient to describe the run-off for some applications.

By definition, run-off is the proportion of precipitation that does not infiltrate. Soil infiltration properties, which are primarily controlled by soil texture, topography (slope) and soil structure (granular vs compact soils), are therefore important (together with the degree of soil sealing) in determining the total amount of surface run-off during precipitation. Similarly, the soil water/moisture content at the onset of precipitation will significantly influence the amount of run-off.

In this study, run-off is calculated for each pixel and each time-step in the design storm precipitation time series. We will consider the net run-off from each pixel at time $t$, $R_t$, as consisting of two components, one arising from impervious areas, $R_{t,IS}$, the other from non-impervious areas, $R_{t,Non-IS}$. This dual approach is necessitated by the relatively coarse spatial resolution (30m) of the Landsat imagery and the large heterogeneity of urban areas, which in many cases causes both pervious and impervious surfaces to be present within individual grid cells:

$$R_t = R_{t,Non-IS} + R_{t,IS} \tag{2}$$

Run-off from the pervious areas is here based on a simplified version of Horton's infiltration model (Horton, 1933). Equation (3) presents Horton's original infiltration model where $f_t$ is the infiltration rate at time $t$, $f_0$ is the initial infiltration capacity, $f_c$ is the constant or equilibrium infiltration rate, and $k$ is the soil-specific decay constant:

$$\boldsymbol{f_t = f_c + (f_0 - f_c)e^{-kt}} \tag{3}$$

Assuming that soils are fully saturated at the onset of the precipitation, which means that the initial loss ($f_o$) is negligible, we can apply the simplified Horton's infiltration equation (4), where the infiltration capacity is constant during individual high-intensity precipitation events. For extreme precipitation events, some authors further report the initial loss to be negligible




for the total infiltration/run-off (Stone et al., 2008). Adopting this simplifying assumption, we can apply the same soil texture data-set across the different cities. In addition, a constant infiltration rate can be used during the entire duration of the modelled precipitation and for all design events. The fact that infiltration capacities are conservative under this assumption may lead run-off from pervious surfaces to be exaggerated in situations where the soils are not saturated at the beginning of

the rainfall event. This may also result in conservative estimates of soil-sealing (e.g. due to urban development). Due to a large variation in reported initial infiltration capacities, and because such values are difficult to extrapolate to other geographical areas, a constant infiltration rate is preferred in the current study. As all grid cells are quantified by their degree of imperviousness, IS values are included to represent the impervious surface fraction for each pixel, causing a decrease in infiltration as the degree of imperviousness increases. Note that $f_t$ equals zero for IS = 100%. This leads to the simplified

infiltration model for pervious areas (IS values are specified as percentages):

$$f_t = f_c (100 - IS) \tag{4}$$

From this simplified infiltration model, the run-off contribution from pervious areas, $R_{t,Non-IS}$, is calculated for each pixel and

time-step using equation (5), where $P_t$ is the precipitation rate and $f_t$ the infiltration rate. IS values are the share (%) of each grid cell covered by impervious surfaces.

$$R_{t,Non-IS} = (P_t - f_t)(1 - IS) \text{ if } P_t > f_t \text{ and } R_{t,Non-IS} = 0 \text{ if } P_t < f_t \tag{5}$$

Run-off from impervious areas is calculated by assuming that there is no infiltration (Jensen 1990, Butler and Davies 2011). However, the presence of a drainage system is simulated by removing run-off corresponding to precipitation from a five-year return period, $P_{t,RP5}$. Hence the contribution to the net run-off from the impervious areas, $R_{t,IS,}$ is given by:

$$R_{t,IS} = (P_t - P_{t,RP5})IS \tag{6}$$

Run-off from grid cells that includes both pervious and impervious areas, $R_{t,\%IS,}$ can be calculated by combining equation (5) and (6) with information on the shares of the two types of surfaces within the individual grid cells:

$$R_{t,\%IS} = (P_t - f_t)(1 - IS) + (P_t - P_{t,RP5})IS \tag{7}$$


The average run-off ratio over the entire precipitation event, $R_r$, is calculated using equation (8), i.e. the share of precipitation which does not infiltrate or is transported via the sewer system. The ratio is calculated prior to the horizontal movements of





water and driven by the characteristics of the DEM in the MIKE21 overland flow model (see section 2.5 for information on the flood modelling).

$$R_r = \frac{\sum R_t}{\sum P_t} \tag{8}$$

In the current study, potential soil infiltration rates are estimated for each city using information on the dominant soil texture and the average slope of the urban area (Figure 3). Soil-texture data at a spatial resolution of 10km*10km from the European Soil Portal are used to provide information on the general soil properties at the city level (JRC, 2016). An average of the four grid cells closest to the cities was converted to a single average soil-texture class. The average slopes were calculated using

10 the EU-DEM, which was also used for the overland flow model. Data from the United States Department of Agriculture (USDA) on soil-type specific infiltration are used as the basis for estimating the potential infiltration rates, as well as to highlight the variations in soil textures and topography between the different cities (USDA, 2016) (Figure 3). High and low values of potential infiltration are included to examine the sensitivity of the modelling approach to variations in this parameter and to provide a quantitative measure of uncertainty to be used in the ensuing analyses of exposure to pluvial

15 flooding.

| Soil texture/slope | | 0-4 % | 5-8 % | 8-12 % | 12-16 % | >16 % |
|---|---|---|---|---|---|---|
| Coarse Sand | Coarse | 32 Odense | 25 | 19 | 13 | 8 |
| Medium Sand | | 27 | 22 | 16 | 11 | 7 |
| Fine Sand | Medium | 24 Vienna | 19 | 14 | 10 | 6 |
| Loamy Sand | | 22 | 18 | 13 | 9 | 6 |
| Sandy Loam | | 19 | 15 | 11 Nice | 8 | 5 |
| Fine Sandy Loam | Medium fine | 16 | 13 | 10 | 6 | 4 |
| V. Fine Sandy Loam | | 15 Strasbourg | 12 | 9 | 6 | 4 |
| Loam | | 14 | 11 | 8 | 6 | 4 |
| Silt Loam | Fine | 13 | 10 | 8 | 5 | 3 |
| Silt | | 11 | 9 | 7 | 5 | 3 |
| Sandy Clay | Very fine | 8 | 6 | 5 | 3 | 2 |
| Clay Loam | | 6 | 5 | 4 | 3 | 2 |
| Silty Clay | | 5 | 4 | 3 | 2 | 1 |
| Clay | | 3 | 3 | 2 | 1 | 1 |

| mm/hr | France | Austria | Denmark | France |
|---|---|---|---|---|
| | Strasbourg | Vienna | Odense | Nice |
| High | 30.6 | 46.2 | 58.6 | 20 |



| Average | 15.3 | 23.1 | 29.3 | 10 |
|---|---|---|---|---|
| Low | 7.65 | 11.55 | 14.65 | 5 |

**Figure 3: Potential infiltration rates (mm/hr) for pervious areas in the four cities, calculated based on the primary soil texture and average slope within the cities. Low = 0.5 * average, high = 2 * average, adapted from USDA, 2016.**

## 2.4 Climate-change impacts on extreme precipitation

Climate change is expected to increase the intensity and frequency of precipitation extremes in both the short and long terms for most regions globally, including Europe (Fowler and Hennessy, 1995; Larsen et al., 2009; Field et al., 2012; Sunyer et al., 2015a). Climate projections generally suggest that the most severe extremes with the shortest (sub-daily) durations are likely to be enhanced more than less severe (daily) extremes (Larsen et al., 2009; Arnbjerg-Nielsen, 2012). In this analysis, we consider two different climate scenarios (Representative Concentration Pathways): RCP 4.5 and RCP 8.5. The RCP 4.5 scenario describes a mitigated future with increases in near-surface air temperatures of 1.8°C (1.1-2.6°C) towards 2100 compared to the present-day reference period, while the RCP 8.5 scenario represents a world where the increased radiative forcing corresponds to an increase of 3.7°C (2.6-4.8°C) in 2100 (Intergovernmental Panel on Climate Change, 2014).

The impact of climate change on extreme precipitation intensities is represented using a change factor (CF) methodology, i.e. by estimating the relative difference between climate model outputs (extreme precipitation intensities) for future and present-day conditions respectively (Willems et al., 2012; Sunyer et al., 2015b). Extreme value analysis is carried out for both present-day (1986-2005) and future (2081-2100) time slices for maximum hourly precipitation (within one day) to estimate the intensities for return periods of five to a hundred years (RP5, RP10, RP20, RP50 and RP100). The extreme value series are derived from the two time slices using a Partial Duration Series (PDS) based on an average of three extreme events per year, following Sunyer et al. (2015b). A Generalised Pareto Distribution (GPD) is then fitted to the derived extreme value series to estimate the intensities for different return periods. In contrast to Sunyer et al. (2015a), the shape parameter in the GPD is not estimated using a regionalisation approach, but calculated individually for each grid cell. This potentially causes a large diversity in resulting CFs between neighbouring grid cells. The CFs are calculated for each return period (RP5-RP100) and applied to the statistic of present-day design precipitation events to obtain the characteristics of future precipitation. The same CF is used for all durations in the Intensity Duration Frequency (IDF) curves.

An ensemble of ten regional climate projections extracted from the CORDEX archive are used to estimate different CFs (Giorgi et al., 2009). All the simulations were carried out by the Swedish Meteorological and Hydrological Institute (SMHI) and use SMHI's regional climate model RCA4 to downscale climate projections from ten different General Circulation Models (GCMs) to a horizontal resolution of fifty km. Three different CFs are estimated for each city and climate scenario: a CF corresponding to the ensemble mean, as well as CFs corresponding to the 10[th] and 90[th] percentiles (based on the ten simulations; 80% of the models are within the 10[th] and 90[th] percentile) to represent some of the model uncertainty. In all



cases the CFs are calculated based on the nine grid cells located closest to the four cities. This is done to avoid selecting individual pixels with abnormally low or high values, which could be present because of the spatially heterogeneous shape parameter in the GPD.

**2.5 Flood modelling**

Overland flooding can occur as a consequence of extreme precipitation when the urban drainage system is surcharging or when soil infiltration and storage capacities are exceeded. Water can pond on the surface or flow in preferential flow paths along streets or between buildings depending on the local topography. When modelling overland flooding, several modelling concepts are available. Some methods integrate 1D modelling of flow in the sewage system with a 1D description of overland flow (Maksimović et al., 2009). Others use modified software originally intended for estuarine and coastal

modelling to interact with the drainage system, e.g. MIKE FLOOD by DHI or TUFLOW (MIKE Powered by DHI, 2016; TUFLOW, 2016). The pros and cons of these methods have been widely discussed (Mark et al., 2004; Leandro et al., 2009; Obermayer et al., 2010), and the choice of method is highly dependent on the aim of the study: e.g. 1D solutions are fast but offer a poor approximation of complex (non-unidirectional) flows.

In this study we use the overland flow model in MIKE 21 (MIKE Powered by DHI, 2016), which computes 2D overland flows in response to given extreme precipitation input based on, e.g., terrain data, including a DEM. The main data input to MIKE 21 for the analyses conducted in this study are time series of precipitation and infiltration rates for the geographical locations being examined, along with a DEM for routing the excess surface water after the onset of precipitation. This modelling approach does not include an explicit representation of subsurface flows, nor of the urban drainage system.

Instead the drainage capacity of the pipes is included by modifying the precipitation input over the entire urban area by assuming a general maximum pipe capacity based on precipitation intensities (Chow et al., 1988; Henonin et al., 2013). Here the maximum drainage capacities are assumed to correspond to precipitation with a return period of five years (RP5). The intensities of precipitation events with return periods exceeding this level are reduced accordingly for the impervious surfaces to reflect this assumption (see equation 6 in section 2.3). The result gives the run-off generated at a pixel level. This

is then routed between pixels to allow for further losses if downstream pixels have surplus infiltration capacity. Effectively this means that the run-off ratio prior to horizontal flows is calculated for each grid cell for every time step (min) using equation 7.

Evidently, by neglecting the exact location and characteristics of the urban drainage system, errors are introduced which

particularly influence the results with respect to the extent and location of flooding derived from the model. However, it is assumed that these effects will be localized and that their significance will decrease for more extreme precipitation events (Paludan et al., 2011).





IDF curves are used to construct time series of design precipitation events for 5, 10, 20, 50 and 100 RPs (Gregersen et al., 2014; BMLFUW, 2016; Coste and Loudet, 1987). The IDF curves are derived from historical measurements of precipitation from weather stations located in (or in the surrounding areas of) each of the cities and are considered to be representative for

precipitation characteristics in both 1984 and 2014. The duration of a precipitation event is set to four hours in all our simulations to ensure that the results are comparable in both time and space. **Error! Reference source not found.** in the supplementary materials shows total precipitation for the different precipitation events and cities under present-day and future climatic conditions. The EU DEM, which offers elevation estimates over the European continent at a 25m spatial resolution, is used as the basis for calculating surface water flows after the onset of precipitation (EEA, 2016). Soil water

infiltration rates are calculated for each grid cell by combining IS fractions with information on soil textures and average slope at the city level (see section 2.3 for more details).

## 3 Results

### 3.1 Urban Development Analysis

The results of the urban development analyses show that the four cities have experienced increasing imperviousness

throughout the past thirty years, which is consistent with the predominant trend towards urbanisation worldwide, with absolute changes ranging from 6.6 % in Nice to 11.6 % in Strasbourg (Table 1). From visual inspection we find that the increases in IS are primarily driven by cities expanding into non-urban areas. However, there is also a tendency towards the intensification of existing urban land cover in all four cities. Detailed quantitative analyses of the location of change are highly relevant for other applications, including risk assessment, but they lie outside the scope of this paper.

|  | France | Austria | Denmark | France |
|---|---|---|---|---|
|  | Strasbourg | Vienna | Odense | Nice |
| 1984 | 41.4 % | 42.2 % | 29.1 % | 38.1 % |
| 2014 | 53.0 % | 53.5 % | 36.6 % | 44.7 % |
| Change | 11.6 % | 11.3 % | 7.5 % | 6.6 % |

**Table 1: Impervious surface fractions and changes therein during 1984-2014 (2013-2015). Imperviousness is calculated for the area covered by the urban classes in the CORINE 2012 Land Cover Inventory (CORINE, 2012).**

### 3.2 Climate-change impacts on extreme precipitation

The results of the extreme value analysis indicate a general increase in the intensity of extreme precipitation towards the end

of the 21st century for both the RCP 4.5 and RCP 8.5 scenarios, with CFs of ≈ 1- 1.6 (Table 2). Average CFs are found to vary less (≈1.1-1.3 for RCP 4.5 and 1.2-1.3 for RCP 8.5), leading to expected intensity increases of 10-30%. As expected, the average CFs are found to increase the most for the most extreme events (RP100), and in the majority of cases (except for




Nice) those for RCP 8.5 are found to be higher than for RCP 4.5. In addition, we observe a large variation in CFs when using the different GCMs, as the 90[th] percentile and 10[th] percentile values differ considerably from the ensemble mean. For some areas, few models project a decrease in the intensity of extreme precipitation (CFs < 1). The variation in CFs between different models confirms the findings of previous research efforts (e.g. Hawkins and Sutton, 2011) that model uncertainty

5  remains a primary source of uncertainty in precipitation projections.

| Parameter | Return period | France | | Austria | | Denmark | | France | |
| | | Strasbourg | | Vienna | | Odense | | Nice | |
| | | RCP45 | RCP85 | RCP45 | RCP85 | RCP45 | RCP85 | RCP45 | RCP85 |
|---|---|---|---|---|---|---|---|---|---|
| 90[th] percentile | RP10 | 1.22 | 1.33 | 1.25 | 1.37 | 1.18 | 1.28 | 1.30 | 1.37 |
| | RP20 | 1.24 | 1.38 | 1.29 | 1.39 | 1.20 | 1.32 | 1.39 | 1.42 |
| | RP50 | 1.32 | 1.46 | 1.40 | 1.43 | 1.26 | 1.39 | 1.53 | 1.51 |
| | RP100 | 1.39 | 1.54 | 1.51 | 1.46 | 1.34 | 1.45 | 1.63 | 1.59 |
| Average | RP10 | 1.13 | 1.26 | 1.12 | 1.25 | 1.07 | 1.17 | 1.18 | 1.19 |
| | RP20 | 1.15 | 1.28 | 1.13 | 1.27 | 1.08 | 1.17 | 1.22 | 1.21 |
| | RP50 | 1.17 | 1.32 | 1.15 | 1.30 | 1.10 | 1.18 | 1.27 | 1.25 |
| | RP100 | 1.20 | 1.35 | 1.18 | 1.33 | 1.12 | 1.20 | 1.32 | 1.29 |
| 10[th] percentile | RP10 | 1.06 | 1.14 | 1.01 | 1.17 | 1.01 | 1.09 | 1.06 | 1.02 |
| | RP20 | 1.04 | 1.12 | 1.00 | 1.14 | 0.98 | 1.08 | 1.03 | 1.00 |
| | RP50 | 1.02 | 1.09 | 0.98 | 1.12 | 0.97 | 1.04 | 1.05 | 1.00 |
| | RP100 | 1.00 | 1.08 | 0.96 | 1.13 | 0.96 | 1.00 | 1.07 | 1.01 |

**Table 2: Change factors for hourly precipitation for the period 2081-2100 (control period: 1986-2005) for RCP 4.5 and RCP 8.5**
10  **for Strasbourg, Vienna, Odense and Nice. Results are based on regional climate projections using the RCA4 regional climate model, downscaling ten different GCMs: CANESM2, CSIRO, CERFACS, ICHEC, IPSL, MIROC, MOHC, MPI, NCC and NOAA (ESGF, 2016).**





### 3.3 Run-off from impervious and pervious areas

Figure 4 shows the run-off ratios for the different cities in 1984 and 2014 (2013-2015) divided into the proportions for impervious ($R_{t,IS}$) and pervious areas ($R_{t,Non-IS}$). As expected, the total run-off ratio is highest for the most extreme events (i.e. RP100) and decreases with the intensity of the precipitation. Regional differences in precipitation characteristics, degree

of imperviousness and soil infiltration properties causes total run-off ratios to vary considerably between the different cities, with present-day run-off ratios ranging from 26-44% for Odense to 52-79% for Nice. In general we find that $R_{t,Non-IS}$ accounts for the largest part of the total run-off, although with differences between different locations and events. Some important observations should be noted. First, we see that run-off from pervious areas ($R_{t,Non-IS}$) is most important for the smallest events, and vice versa for impervious areas ($R_{t,IS}$). Secondly, regional differences in imperviousness and especially

soil infiltration causes $R_{t,Non-IS}$ to dominate run-off for Nice, while the shares are comparable for Odense, Vienna and Strasbourg. Finally, an increase in run-off ratios during 1984-2014 can be identified for all four cities consistent with the increase in the IS fractions also found in this period (Table 1). Not surprisingly, we find that $R_{t,IS}$ increased and $R_{t,Non-IS}$ decreased during 1984-2014 (2013-2015). Overall we find the amount of run-off from pervious urban areas to be considerable, suggesting that in many cases future precipitation intensities lead to higher amounts of surface run-off than can

be infiltrated into the soils.

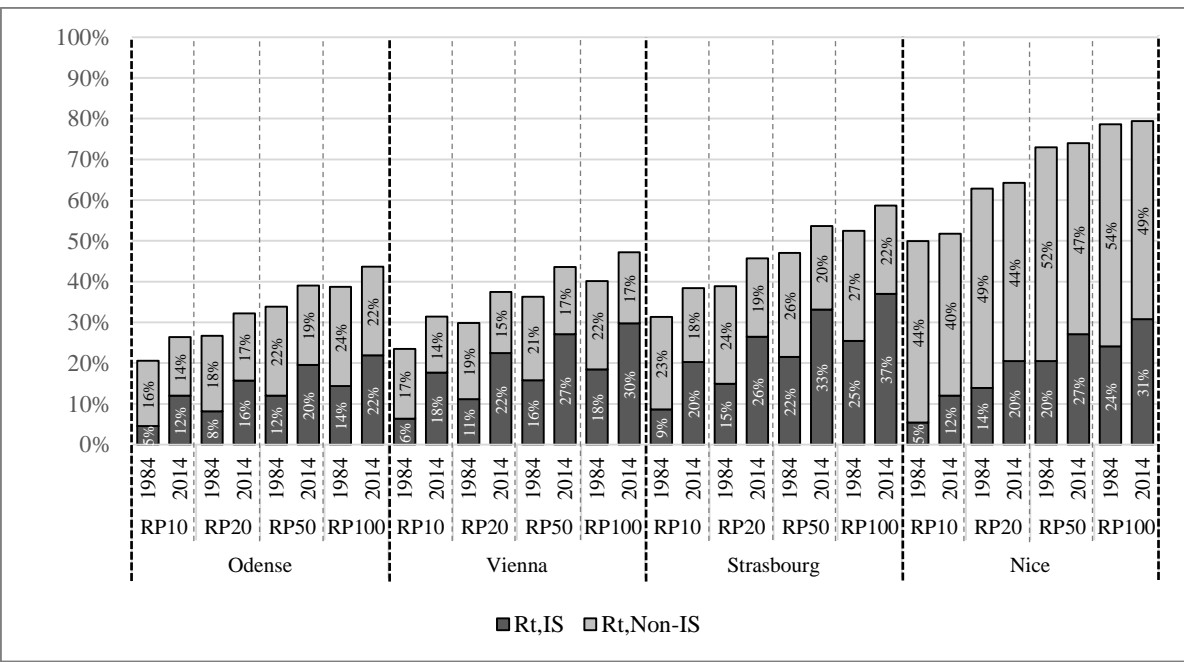

**Figure 4: Run-off ratios from urban areas for precipitation events of different intensities, RP10-RP100, divided into shares from impervious ($R_{t,IS}$) and pervious surfaces ($R_{t,Non-IS}$). Run-off ratios are shown for historical (1984) present-day imperviousness (2014 – drainage system as in 1984) and using present-day precipitation intensities.**





## 3.4 Impacts of urban development and climate-change on pluvial flooding

The primary outputs from the flood-model simulations are in the form of flood-hazard maps showing the maximum flood depth and extent for each individual simulation (Figure 5). We will quantify the impacts of urban development and climate change through a cross-comparison of multiple flood-hazard maps.

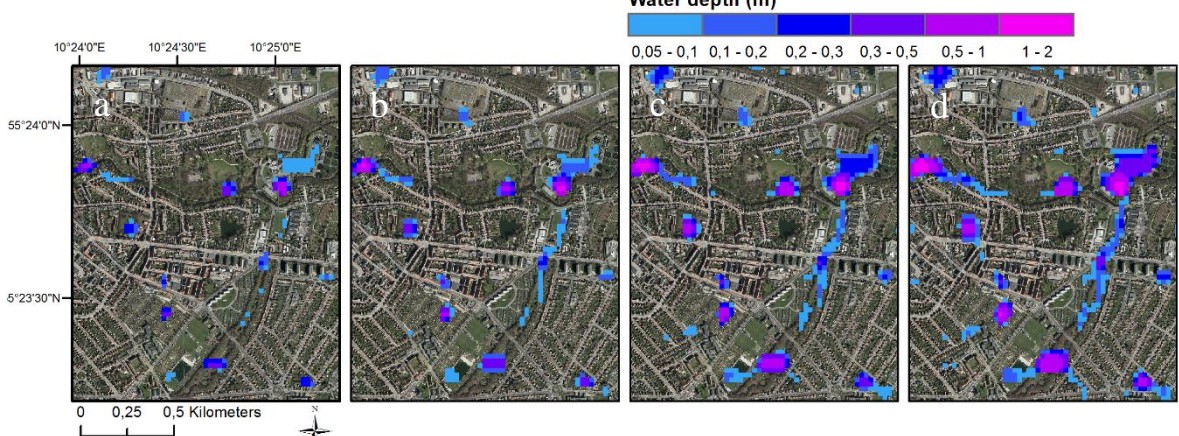

**Figure 5: Flood-hazard maps for sub-area in Odense showing the maximum water depth and extent during extreme precipitation events with return periods of (a) 10, (b) 20, (c) 50 and (d) 100 years for RCP 8.5 in the period 2081-2100 using average climate factors and present-day imperviousness.**

The results of the combined remote-sensing and flood-modelling analyses (Figure 6) depict a clear pattern of increased flooding for events with longer return periods and for events simulated with elevated levels of imperviousness, the latter due to urban development during 1984-2014. Flooding is defined here as occurring when maximum surface water depths exceed 10 cm. Similarly, the effects of climate change are found to increase pluvial flooding substantially for all return periods and for both climate scenarios, with RCP 8.5 having a larger impact in most cases. Only in the case of Nice are the results for RCP 4.5 and RCP 8.5 comparable. Uncertainties in the estimated impacts of urban development and climate change are examined by varying, respectively, potential soil infiltration rates and climate factors for each of the four cities and are represented as error bars in all the following figures. The impacts of uncertainties related to the projection of future precipitation intensities (i.e. climate change) are comparable to those of accurately estimating soil-infiltration properties (i.e. urban development) (Figure 6). However, there is a tendency towards a greater influence of uncertainties in soil infiltration for the smallest events and vice versa for precipitation with the highest intensities. In addition, in all cases except for Nice we observe that the uncertainty increases with the intensity of precipitation, and the error bars widen when we move towards the most extreme events (Figure 6). There are notable differences in how severely the urban areas are impacted by flooding depending on their geographical location, climate conditions, soil properties and topographies. For Odense, the observed share of total area prone to pluvial flooding approximates to 5% for the most severe events while approaching 10% for Vienna and Strasbourg and 20% for Nice. The results of the flood-model simulations where the drainage system is updated to follow urban development (2014d in Figure 6) and climate change (RCP45d and RCP85d in Figure 6) suggest several



important implications. First, developing the urban drainage system in relation to increases in imperviousness (i.e. due to urban development) appears to compensate fully for the increase in flooding caused by the additional run-off from sealed surfaces. In contrast, updating the drainage system according to increases in precipitation intensities (i.e. caused by climate change) only marginally reduce flooding for the largest events, primarily because the increased capacity is small when
compared to both the current and future changes in the intensity of these very severe events.

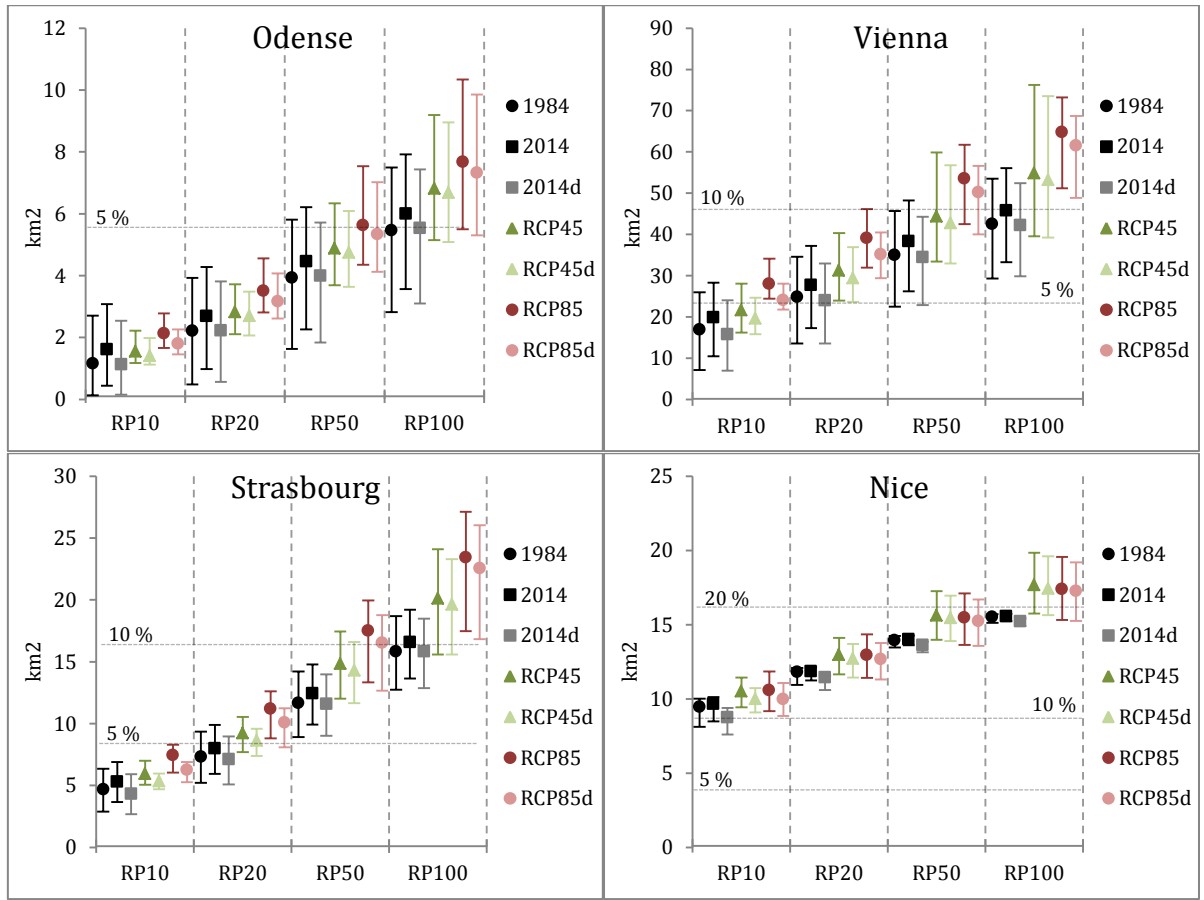

**Figure 6: Total area flooded by more than 10 cm of surface water as a result of extreme precipitation events of different return periods (RP10-RP100); the years indicate the observed configurations of impervious surfaces in 1984 and 2014 respectively. 2014d**
**= drainage system updated to follow changes in imperviousness caused by urban development. RCP45d and RPC85d = drainage system updated to follow changes in precipitation intensities caused by climate change. Error bars represent low/high infiltration rates and low/high climate factors (CFs) respectively (low CF = 10th percentile, high CF = 90th percentile). The horizontal %-lines highlight the share of the total urban area in each city.**

In comparing the absolute changes in exposure to flooding caused by urban development and projected climate change
respectively, we find that the relative influence of these two drivers varies considerably between the different cities (Figure 7). To compare the influence of thirty years of urban development in 1984-2014 with climate change in a 100-year perspective (1986-2005 to 2081-2100) the results are normalised and shown in terms of 'change per year'. For Odense, Vienna and Strasbourg, the influence of recent urbanisation is comparable to that of increased precipitation extremes under



the RCP 8.5 (Odense) and RCP 4.5 scenarios (Vienna and Strasbourg), whereas climate change is the most important driver for Nice. This is consistent with the variations in the properties of soil infiltration between the four cities (Figure 3). The relative importance of climate change increases in all cases when moving towards the most extreme events (i.e. RP50 and RP100). We observe a large uncertainty related to the climate change projections, especially for the most extreme events,

with a few models even projecting a decrease in precipitation intensities and in overall flood exposure for Vienna and Odense (error bars below 0 for RP20-RP100). For Odense, Vienna and Strasbourg, we find that the absolute changes in pluvial flooding increase with the intensity of the events for both urban development and climate change, whereas the same trend is not observed for Nice. This difference could be explained by the presence of extensive topography (average slope of 8%) within Nice, causing surface waters to flow over greater distances. As a result, changes to IS that occur outside the

urban zone may to some degree influence the simulated flooding within this area. Also, the findings imply that continued urban development over a 100-year period could potentially facilitate an increase in flooded areas corresponding to approximately 1-2% of the total urban area (0.01%/year – horizontal lines), increasing to 5% for some cities (0.05%/year – horizontal lines) as a consequence of high-end climate change (RCP 8.5). The results presented in Figure 7 confirm that updating the drainage system in relation to increases in imperviousness is an efficient strategy to mitigate the adverse effect

of soil-sealing (no increase in flooding for *UDd* in Figure 7). Conversely, developing the drainage system according to the impact of climate change on precipitation intensities only marginally reduces flooding (RCP45d and RCP85d in Figure 7).

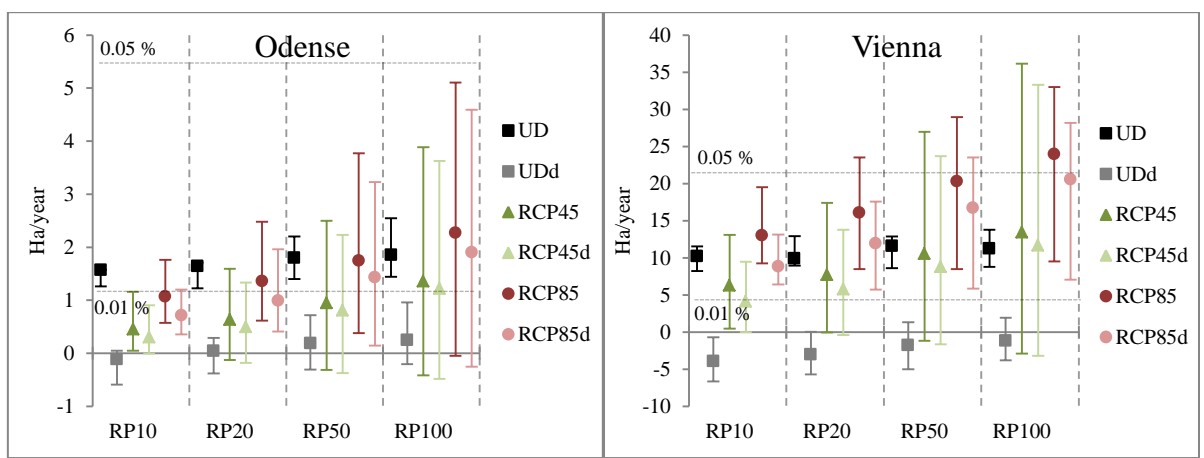





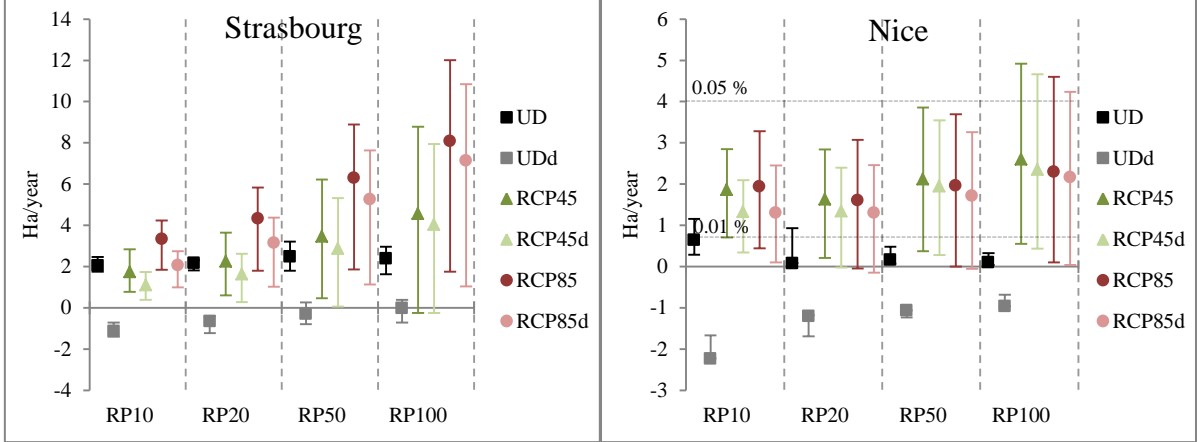

**Figure 7: Averaged annual change in the total flooded area (>10cm water depth) due to the impacts of urban development (UD) and climate change (RCP 4.5 and RCP 8.5) on extreme precipitation for different return periods (RP10-RP100). UDd = drainage system updated to follow changes in imperviousness caused by urban development. RCP45d and RPC85d = drainage system updated to follow changes in precipitation intensities caused by climate change. Error bars represent low/high infiltration rates and low/high climate factors (CFs) respectively (low CF = 10th percentile, high CF = 90th percentile). The horizontal %-lines highlight the share of total urban area for each city.**

Lastly, we have assessed the sensitivity of our results to the selected threshold (10 cm), which is often defined as the surface water depth where damage begins to occur. The results are normalized to indicate the average change in flooded area every time absolute imperviousness increases by 1% at the city scale. We explore the influence of urban development on flood exposure for four different thresholds, ranging from 1 to 20 cm (Figure 8). First, we observe that the estimated effect of urban development increases more for higher thresholds (20 cm) as compared to lower flood depths. This relationship is generally present for all cities and precipitation events. A small variation in flood response is observed in most cases, confirming that the findings are only marginally sensitive to the choice of flood threshold. The large increase in flooded area for the 20cm threshold (>10%) in the case of RP10 for Odense is due to a very small area being affected by flooding at high flood depths for this event, and thus a high predisposition for experiencing a significant relative change in the case of increasing imperviousness. Secondly, we find that the influence of urban development decreases rapidly for all flood thresholds as we move towards the most extreme precipitation events. In addition, we find that the relative importance of urban development and climate change is unaffected by variations in flood thresholds, indicating that our findings are robust to the selection of this parameter.





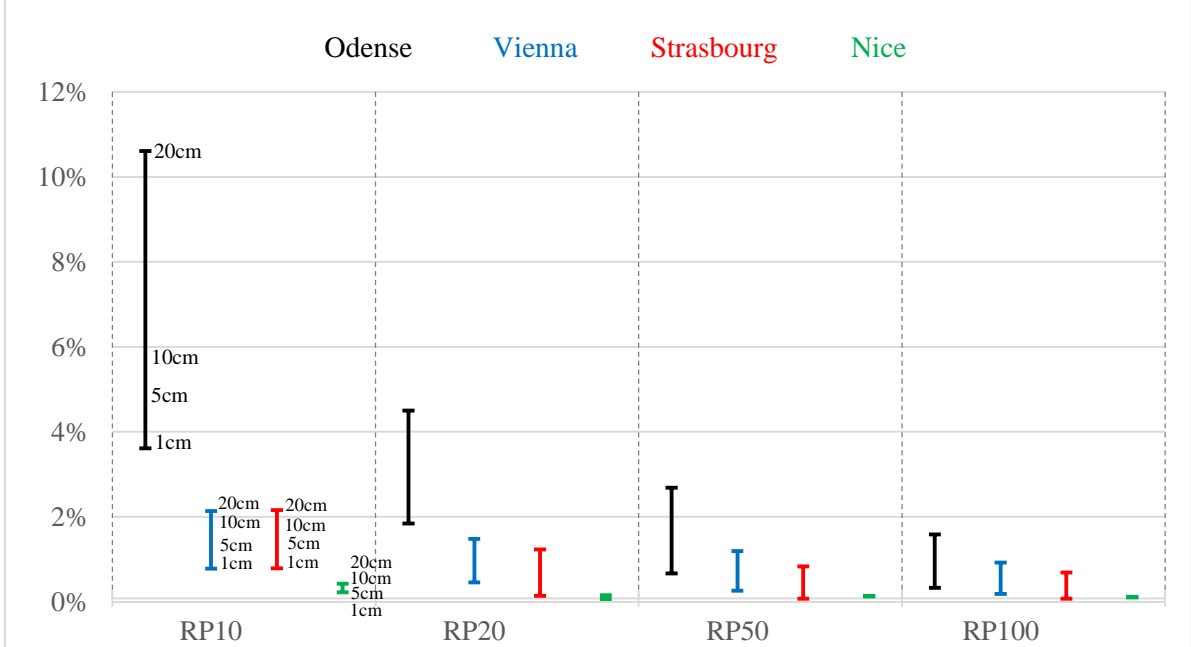

**Figure 8: Relative change in flooded area per 1% increase in imperviousness. Results are shown as the range of change for water depths of 1, 5, 10 and 20 cm.**

## 4 Discussion

5   Assuming that the soil is fully saturated at the onset of the precipitation and that the initial loss is negligible, we applied a simplified Horton's infiltration model (3), where infiltration capacity is constant during precipitation. This scenario assumes conservative infiltration capacities and may exaggerate run-off from pervious surfaces in situations where the soils are not saturated at the onset of precipitation. Similarly, this may also result in conservative impacts of soil-sealing (i.e. due to urban development). Due to a large variation in reported initial infiltration capacities, and because such values are difficult to 10  extrapolate to other geographical areas, a constant infiltration rate is preferred in the current study. In addition, initial losses have previously been found to be negligible for the total infiltration/run-off during extreme precipitation (Stone et al., 2008), which is the type of precipitation considered in the present study.

Climate-change impacts on future extreme precipitation (and consequently on pluvial flooding) and the influence of urban 15  development in exposing cities to floods are both surrounded by large uncertainties. The primary uncertainties in the projection of future precipitation extremes can be attributed to the incomplete understanding of processes and components in the Earth System, resulting in large model uncertainties and thus large variations in projected change factors between different models (Hawkins and Sutton, 2011; Sunyer et al., 2015a). A diversity of downscaling methods, climate scenarios and natural internal variability, especially in the short term, also contribute to a high level of uncertainty when projecting





precipitation extremes. In the current study, the uncertainty related to climate change is addressed by including different climate scenarios, as well as high/low climate change factors, corresponding to the 90[th] and 10[th] percentiles, from a set of ten different GCMs. The resulting change factors derived for future precipitation were found to be in close agreement with the findings of other studies (Larsen et al., 2009; Willems et al, 2012; Sunyer et al., 2015a), although direct comparisons are limited by differences in key variables and methodological assumptions between these studies. These include the choice of climate scenarios, control and scenario periods and the characteristics of the extreme events. It is generally agreed that change factors and related uncertainties are elevated for precipitation events with longer return periods and for higher-end climate scenarios (i.e. larger change factors for RP100 and RCP 8.5), and our findings confirm these patterns. On the basis of analyses of output from one regional climate model, Larsen et al., (2009) found change factors of 1.27, 1.35 (France), 1.19, 1.21 (Austria) and 1.36, 1.60 (Denmark) for RP20 and RP100 respectively at the end of the 21th century under the IPCC A2 scenario. These factors are similar to what was observed for the RCP 8.5 scenario in the current study (Table 2).

The observed increase in flood exposure as a consequence of urban development (i.e. changes to impervious surfaces) is comparable with the findings of previous studies (Poelmans et al., 2010; Dams et al., 2013; Verbeiren et al., 2013). In these studies, recent increases in urban footprints in north-west Europe are found to produce similar elevated run-off quantities at a catchment scale during both normal and extreme precipitation. The above authors find exaggerated run-off volumes of 1.5%- 2.5% to be expected for a change in urban land cover/imperviousness of 1%, in close agreement with the findings of the current study (Figure 8). Also, the observed decline in the importance of soil-sealing when moving towards more extreme precipitation events confirms the findings of Hollis (1979), who concludes that changes to imperviousness are negligible for flood magnitudes with return periods of > 100 years, while causing discharge rates to increase as much as tenfold for very frequent events (return periods of < 1 year).

Knowledge about the individual and combined effects of historical urban development patterns, expected future precipitation and the ensuing changes to large-scale (city-level) exposure to flooding can assist decision-makers and city planners to develop climate resilient cities. For example, from our analysis we find that urban development in Odense and Vienna influences the extent of flooding considerably, while only marginally affecting the degree of flooding for Strasbourg and Nice (Figure 7). This suggests that further soil-sealing in Odense and Vienna (and similar urban areas) should be considered very carefully, as it may substantially increase their exposure to pluvial flooding. In addition, we observe a decline in the impact of soil-sealing as precipitation intensities increase (greater impacts for shorter return periods; see Figure 8), which implies that using green urban areas as adaptation measures is most efficient for the least severe events, while not providing any noticeable protection against flooding due to high-intensity precipitation. We find that updating the drainage system to correspond to changes in imperviousness, as is common in many countries, completely mitigates the adverse impacts of soil-sealing on flooding. This clearly indicates that current practices are often sufficiently effective as large-scale adaptation measures in addressing the impacts of urban development on pluvial flooding. Following a strategy where the design of the



drainage system is specifically updated to match that of expected increases in precipitation intensities caused by climate change mainly affects the frequency of pluvial flooding while only marginally reducing the amount of flooding for the largest events. This is partly caused by the positive correlation between CFs and precipitation intensities (Table 2), which leads to a relatively large increase in the intensity of very extreme events (RP20-RP100), as compared to the smaller events
(RP5), from which the drainage system is designed.

In terms of increased flood risk, the effect of the last ~30 years of urban development in several of the cases is found to be comparable to the impacts of a moderate climate-change scenario, i.e. RCP 4.5. In adapting an optimistic mind-set when interpreting the results of our analysis, one may therefore argue that climate change may potentially not influence flooding to
an extent that far exceeds what we are already used to dealing with in relation to urban development. We may just need to address the challenge a bit differently from what we have been used to. The four cities in the analyses represent a wide range of soil infiltration rates, which is the key driver determining the impact of soil-sealing (e.g. caused by urban development), and the reported impacts of urban development on flood severity are expected to be representative for many other cities, in particularly in Europe. This indicates not only that should future city development occur where flood risk is negligible, but
also that the retrofitting of existing areas of cities should receive great attention when planning future climate-resilient cities. Evidently, the relative importance of urban development and climate change for overall exposure to flooding will differ according to current and future city- and region-specific urbanisation rates. Thus, the results presented here should not be considered valid for regions that are characterized by rapid urbanisation, including some Asian and African megacities. For such regions, urban development will most likely be the primary driver of changes in the exposure of cities to pluvial
flooding.

The results of the urban development analysis should be considered fairly robust, as the method applied here is well established. However, a few aspects deserve additional attention. First of all, the remote-sensing technique used to estimate imperviousness and changes in this study is only strictly applicable within urban areas and is most accurate for cities where
bare soil does not occupy a large proportion of the urban surface. The accuracy decreases considerably for other land-use types (e.g. agricultural areas, forest etc.). Indeed, for surrounding land-cover types, including agricultural areas, the method is likely to introduce large errors. For the four cities considered here, visual inspection confirmed this to be a minor issue. Since urban boundaries rarely coincide with the physical boundaries of watersheds, the modelled area is extended considerably from the urban coverage. A consequence of this is that (wrongly) measured changes in imperviousness outside
the cities could have been the cause of slightly increased or reduced flooding within the urban areas. We expect this to be most pronounced for areas characterized by large elevation differences within short distances, prompting surface water to move over larger distances. Also, some uncertainty remains due to differences in the spectral response function between the Landsat 5 and Landsat 8 sensors, as official conversion factors are not yet available for all locations and environments.



In many instances our simplified modelling approach will be unable to determine locations of flooding with a sufficient level of accuracy. This is partly because no drainage system model was included in the flood model, and also because of the rather coarse resolution (25m grid cells) of the elevation model, which does not adequately represent the actual characteristics of the topography. The assumption of a perfectly designed and maintained urban drainage system with capacities corresponding to a RP5 for all locations is not valid in practice and contributes further to the uncertainty. Evidently, by neglecting the exact location and characteristics of the urban drainage system, errors are introduced which particularly influence the results with respect to the location of flooding as simulated by the model. However, it is assumed that these effects will be localized and that their significance will decrease for more extreme precipitation events (Paludan et al., 2011).

## 5 Conclusions

In this study, we examine the impacts of recent urban development and future climate change in exposing urban areas to pluvial flooding for four cities in Europe using a combined remote-sensing and flood-modelling methodology. Urban development is calculated as changes in imperviousness during 1984-2014 using information on vegetation cover based on data from the Landsat 5 and 8 sensors. Climate-change impacts on extreme precipitation (and related flooding) are quantified using a change factor methodology, and extreme values analyses are conducted for present-day (1986-2005) and future (2081-2100) precipitation time series for two climate change scenarios. We find that the impacts of urban development and climate change are comparable, although with large geographical differences. Urban development is found to have a large influence on flood exposure for urban areas characterized by coarse soil textures and limited topography, as soil infiltration rates are excessive here, causing the influence of soil-sealing to be high in such areas. For Odense and Vienna, the impacts of the changes in precipitation intensities for flood exposure under the RCP 8.5 scenario is of the same order of magnitude as that caused by urban development, while climate change is the dominant driver in exposing Nice and Strasbourg to pluvial flooding for most events. We find flooding to increase by 0-10% for every increase in absolute imperviousness of 1%. We also find that the projection of extreme precipitation is surrounded by considerable uncertainties with large inter-model variation in the estimated change factors. The results show a clear trend towards increased impact of soil-sealing for the least severe precipitation events, while only marginally affecting flooding during precipitation with long return periods. Updating the drainage system according to changes in urban land cover, as is common in many countries, is found to mitigate the adverse impacts of urban development in exposing cities to pluvial flooding. However, upgrading the sewer system to maintain a fixed frequency of flooding only has a minor influence on the amount of flooding for very long return periods.

## 6 Acknowledgements

The authors thank Jakob Luchner and Nina Donna Domingo of DHI for assisting with the extreme value analysis and for technical support in relation to constructing and running the MIKE 21 overland flow model.





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
