# Peer review of "Comparison of the impacts of urban development and climate change in exposing European cities to pluvial flooding"

_Hydrology and Earth System Sciences, 2017_

## Referee Comment (RC1) · Anonymous Referee #1 · 29 May 2017

The authors conducted an interesting modeling experiment and presented a useful conclusion that updating drainage systems in response to urbanization is more effecient that updating in response to climate change. I recommend that the authors address the following technical comments:

Page 12: when describing the changes in imperviousness, it should be expressed as x% points, not just x%. They talk about the changes in values expressed as percent, so x% points are appropriate.

Page 12 line 25: "for some areas, few models project a decrease in the intensity of extreme precipitation." –> you could make it more specific

[Figure]

Page 15 line 12-13:" The impacts of uncertainties....(i.e. urban development)" –> I had a hard time figuring out

---

## Referee Comment (RC2) · S. Thorndahl (Referee) · 1 Jun 2017

The authors present a novel approach to compare the impacts on urbanization and climate change on pluvial flooding in four European cities.

The paper is well written and easily read and understood. Results are interesting in that sense that the impact of urbanization and climate change on flooding is in the same order of magnitude.

There are some assumptions, which in my opinion need to be clarified in order to justify the simplifications of models and inputs. However since focusing on relative comparisons between the four cities and different climate scenarios, the effect of these

assumptions on the final results and conclusions might be small - however still relevant to discuss from a scientific point of view.

Comments

In the abstract and the introduction the presence and impacts of an urban drainage system in terms of pluvial flooding is not mentioned. The first pages would benefit from a clarification on this.

In section 2.1 on the framework, it would be worth noting which type of precipitation input is used in the modelling concept.

Line 1, page 6: Why the near-linear relationship. In eq.1 the relationship is linear.

I think the assumption on assuming fully saturated soils at all times, thus simplifying Horton, might need some more clarification. If you have a dry soil the initial infiltration capacity might easily be a decade larger than the end infiltration. In that case you risk overestimating the surface runoff. For some of the sandy soils, e.g. the example for Odense, has a saturated infiltration capacity of $\sim$30 mm/h. If the initial capacity is ten times larges, it is doubtful that the soil will ever get saturated, since you would need rainfall intensities larger than 300 mm/h (for a longer period). But as you write in the discussion the problem decrease for larger return periods, thus larger rainfall intensities.

In the "method section", page 8, the assumption of subtracting the rainfall intensities with a 5 yr return period needs more clarification. It is discussed later on, but it could be relevant to discuss here also. I think neglecting the drainage system, by subtracting the rainfall from a 5 yr return period might be too simple an approach. Partly due to the fact that not all parts of the drainage systems might be designed for a 5 yr return period, and partly due to the fact that even designed for a 5 yr return period, there might be lots of local capacity left in the drainage system, depending on the rainfall dynamics. Furthermore, what about capacity of bassins, channels, recieving waters, etc.?

[Figure]

Page 12, top. The construction of precipitation events needs to be detailed. Are you using design storms e.g. the Chicago design storm constructed from the IDF-curves? These types of storms require a very linear rainfall- flood response in order to be valid. Has this been investigated? Also, why limit the duration to 4 hours – you might underestimate the role of storage bassins, and natural waterways, only looking at the very short durations, thus high intensity, rainfall.

Page 12. You state that you include the total rainfall amounts in the supplementary material, but I think it would be relevant to present here along with the max. intensities over different durations. In that case it would be possible to compare the infiltration rates to rainfall intensities.

Section 3.4. I guess that you assume all surfaces to be impervious during the flood, meaning that you only account for the infiltrating water to the soil in the rainfall input to the model. In reality you might have a flood where water flows from impervious surface to pervious surfaces and infiltrated, but a I guess this is not accounted for. Please comment on this.

Line 12, page 15: I think the limit of 10 cm water levels considered flooding needs some clarification. If you have grid cells of 25 x 25 m2 and with a min water level of 10 cm you discard 62.5 m3 of water. This is a significant amount! Please comment.

In fig. 6 you use km2 and in fig. 7 you use ha. Please apply same units.

---

## Author Comment (AC1) · 10 Jul 2017

The authors thank the reviewer for the positive and constructive comments, which will assist us to further improve the paper. Find the author responses to the specific review comments in the following.

1. Page 12: when describing the changes in imperviousness, it should be expressed as x% points, not just x%. They talk about the changes in values expressed as percent, so x% points are appropriate.

Author response: This will be changed accordingly

2. Page 12 line 25: "for some areas, few models project a decrease in the intensity of extreme precipitation." -> you could make it more specific.

Author response: This will be clarified following the reviewer's recommendation. The following text will be added to Page 12, line 24 in a revised version of the manuscript: For Vienna and Odense, few models project a decrease in the intensity of extreme precipitation (CFs < 1) for the RCP 4.5 scenario.

3. Page 15 line 12-13:" The impacts of uncertainties....(i.e. urban development)" -> I had a hard time figuring out.

Author response: We thank the reviewer for pointing out that this sentence is a bit unclear. What is meant here is that the impact of uncertainties in urban development on flooding, that is, changes in impervious surfaces, is quantified by simulating the occurrence of the precipitation events using different assumptions about soil infiltration rates. In practice, we are using three different estimates of soil infiltration rates (average/low/high) for the individual cities as a means to visualize the potential uncertainty of soil sealing (i.e. in the context of this paper urban development is defined as changes in impervious surfaces). The text will be modified in a revised version of the manuscript to make this clearer.

---

## Author Comment (AC2) · 10 Jul 2017

The authors thank the reviewer for the positive and constructive comments, which will assist us to further improve the paper. Find the author responses to the specific review comments in the following.

1. The authors present a novel approach to compare the impacts on urbanization and climate change on pluvial flooding in four European cities. The paper is well written and easily read and understood. Results are interesting in that sense that the impact of urbanization and climate change on flooding is in the same order of magnitude. There are some assumptions, which in my opinion need to be clarified in order to justify the

simplifications of models and inputs. However since focusing on relative comparisons between the four cities and different climate scenarios, the effect of these assumptions on the final results and conclusions might be small - however still relevant to discuss from a scientific point of view.

Author response: The authors agrees with the reviewer that some of the assumptions made in this paper would have been too simplistic for other applications, including for detailed planning of drainage system updates or for recommendations for location-specific adaptation measures. However, as the reviewer also notices, since the focus of the paper is on the relative comparison of the impacts of urban development and climate change on flood exposure between four European cities, the effect of these assumptions are not considered to affect the results, nor the conclusions, considerably. The authors agree with the reviewer that the importance and influence of some of the assumptions are highly relevant to discuss in a broader scientific context. In the following we will initiate this discussion, and, where appropriate, amend the manuscript accordingly. However, in order to preserve the focus on the paper we will focus on highlighting what assumptions are reasonable in the context of the study and avoid making (too many) diversions on other possible framings.

2. In the abstract and the introduction the presence and impacts of an urban drainage system in terms of pluvial flooding is not mentioned. The first pages would benefit from a clarification on this.

Author response: The authors agree with the reviewer that the paper will improve by including this information in the initial sections of the paper. A clarification on the presence and impacts of an urban drainage system in terms of pluvial flooding will be included in both the abstract and the introduction. See the following suggestions for insertions:

The following text will be added to a revised version of the manuscript:

Abstract page 1 line 24: In addition, two different assumptions are examined with re-
gards to the development of the capacity of the urban drainage system in response to urban development and climate change. In the "stationary" approach, the capacity resemble present-day design, while it is updated in the "evolutionary" approach to correspond to changes in imperviousness and precipitation intensities due to urban development and climate change respectively.

Abstract page 2 line 4: Developing the capacity of the urban drainage system in relation to urban development is found to be an effective adaptation measure as it fully compensates for the increase in run-off caused by additional sealed surfaces. On the other hand, updating the drainage system according to changes in precipitations intensities caused by climate change only marginally reduce flooding for the most extreme events.

Introduction page 3 line 33: We investigate the effectiveness of updating the urban drainage system in response to urban development and climate change simulating two scenarios; (1) the capacity of the drainage system is updated to correspond to changes in impervious surfaces and precipitation intensities and (2) no modifications in the capacity of the drainage systems are assumed.

3. In section 2.1 on the framework, it would be worth noting which type of precipitation input is used in the modelling concept.

Author response: This information will be included in section 2.1. The following text will be added to a revised version of the manuscript: Page 4, Line 29: High-intensity precipitation events with intensities corresponding to RPs of 10, 20, 50 and 100 years are included in the analysis.

4. Line 1, page 6: Why the near-linear relationship. In eq.1 the relationship is linear.

Author response: This is an error. The text will be changed from "near-linear" to "linear".

5. I think the assumption on assuming fully saturated soils at all times, thus simplifying Horton, might need some more clarification. If you have a dry soil the initial infiltration

capacity might easily be a decade larger than the end infiltration. In that case you risk overestimating the surface runoff. For some of the sandy soils, e.g. the example for Odense, has a saturated infiltration capacity of âĹij30 mm/h. If the initial capacity is ten times larges, it is doubtful that the soil will ever get saturated, since you would need rainfall intensities larger than 300 mm/h (for a longer period). But as you write in the discussion the problem decrease for larger return periods, thus larger rainfall intensities.

Author response: The authors agrees that this assumption may lead to overestimations of surface run-off and consequently exaggerated flood extents. Indeed, based on the work we present here we have made an analysis of typical soil and precipitation characteristics for Denmark and have found that in general the initial conditions will lead to changes in the runoff of approximately 1-6 mm depending on soil type and independent of return period (results are not published yet). However, since the assumption is consistent across all sites, soil types, and return periods, and because we are primarily interested in comparing the influence of urban development and climate change on changes in flooding between the different urban areas, this assumption is not expected to influence the results. Instead, if the purpose of our analyses were to accurately estimate the precise area experiencing flooding during specific precipitation, a more detailed representation of infiltration processing, e.g. including initial losses should be applied. In addition, it should be noted that because infiltration varies considerably across small distances, infiltration data is associated with large uncertainties, and accurate location-specific infiltration data is thus not readily available for many locations. In our analyses, we attempt to address this uncertainly by simulating the precipitation events using low, average and high estimates of infiltration. We discuss this on page 8, lines 7-18. Given that we have not made the analysis of the importance of this assumption the word 'slightly' will be added to line 14 in a revised version of the manuscript.

6. In the "method section", page 8, the assumption of subtracting the rainfall intensities

with a 5 yr return period needs more clarification. It is discussed later on, but it could be relevant to discuss here also. I think neglecting the drainage system, by subtracting the rainfall from a 5 yr return period might be too simple an approach. Partly due to the fact that not all parts of the drainage systems might be designed for a 5 yr return period, and partly due to the fact that even designed for a 5 yr return period, there might be lots of local capacity left in the drainage system, depending on the rainfall dynamics. Furthermore, what about capacity of bassins, channels, recieving waters, etc.?

Author response:We agree that a more complex model would yield more precise results if we collected the data needed to build and calibrate a full 1D2D model of both subsurface infrastructure, topography, and natural waterways. As mentioned in the beginning of this rebuttal, that might have been a reasonable approach if the objective of the study was to make detailed designs of adaptation measures. However, for the purpose of the study the methodology is perfectly valid and has been used before, e.g. (Arnbjerg-Nielsen & Fleischer, 2009) and is recommended as one of several simulation methods for both planning and real-time applications by (Henonin, Russo, Mark, & Gourbesville, 2013). The last, but also important, reason is that we do not have access to the data needed in order to make a 1D2D simulation more accurate than the method we have employed.

7. Page 12, top. The construction of precipitation events needs to be detailed. Are you using design storms e.g. the Chicago design storm constructed from the IDF-curves? These types of storms require a very linear rainfall- flood response in order to be valid. Has this been investigated?

Author response: We do employ CDS-storms and have clarified this in the revised manuscript. We need this assumption first of all because we only have IDF-curves available at some of the sites we study. Our results are not a linear function of neither return period nor rainfall volume so we do not understand this part of the reviewers comment. The key assumption for CDS-storms is that a well-defined time of concentration exists, which we need to assume and think that this is reasonable in the current

situation. Testing of this particular hypothesis was the focus of a presentation at the International Conference on Flood Resilience in Exeter (Newton et al, 2013). Because the proceedings only contain extended abstracts we have been in contact with the authors regarding more detailed reporting of the results. Unfortunately, they have not published these results. The following text will be added to page 12, line 16 in a revised version of the manuscript: ". . . Chicago design storm. . .".

8. Also, why limit the duration to 4 hours – you might underestimate the role of storage bassins, and natural waterways, only looking at the very short durations, thus high intensity, rainfall.

Author response: See also response to point no 6. We have highlighted that we only focus on pluvial flooding, i.e. assumed that the response is so short that flooding from larger waterways can be neglected.

9. Page 12. You state that you include the total rainfall amounts in the supplementary material, but I think it would be relevant to present here along with the max. intensities over different durations. In that case it would be possible to compare the infiltration rates to rainfall intensities.

Author response: We agree with the point of not only presenting the infiltration rates and climate factors but also the actual rainfall intensities. In a revised version of the manuscript we will therefore add the maximum 1 hour intensity as well as total volume of the CDS storms for all return periods and all sites.

10. Section 3.4. I guess that you assume all surfaces to be impervious during the flood, meaning that you only account for the infiltrating water to the soil in the rainfall input to the model. In reality you might have a flood where water flows from impervious surface to pervious surfaces and infiltrated, but a I guess this is not accounted for. Please comment on this.

Author response: Infiltration continues during the flood, meaning that water flowing

from fully saturated surfaces to areas with excess capacity will infiltrate accordingly. To accommodate this, a net infiltration rate is defined in MIKE 21 (infiltration is accounted for as a separate input to the model, i.e. as an evaporation type2 file in MIKE 21 and not as part of the rainfall input) allowing for both precipitation and runoff to infiltrate at any stage during the events. We will clarify this in a revised version of the paper. We will clarify this on page 9, lines 7-8 in a revised version of the manuscript by adding the following text: The runoff from each grid cell is allowed to infiltrate later during the events, i.e. runoff flowing from fully saturated surfaces to areas with excess capacity during the events will infiltrate accordingly. To accommodate this, a net infiltration rate is defined in MIKE 21 allowing for both precipitation and runoff to infiltrate at any time during the events.

11. Line 12, page 15: I think the limit of 10 cm water levels considered flooding needs some clarification. If you have grid cells of 25 x 25 m2 and with a min water level of 10 cm you discard 62.5 m3 of water. This is a significant amount! Please comment.

Author response: We have studied several thresholds ranging from 1-50cm (not included in the paper) and found that our results are consistent independent of the selected threshold. Again, see response to point 6: our main focus is not on detailed calculation of the spatial distribution of the floods, but a comparative analysis of the drivers causing changes in flood hazards at city level.

12. In fig. 6 you use km2 and in fig. 7 you use ha. Please apply same units.

Author response: Using the same units leads to very large or very small numbers in one of the graphs. In a revised version of the manuscript we will therefore add the following text to the fig.7 legend: Please note the differences in unit on the y-axis compared to Figure X to each of the figures.

Litterature refered to used in the author responses:

Arnbjerg-Nielsen, K., & Fleischer, H. (2009). Feasible adaptation strategies for increased risk of flooding in cities due to climate change. Water Science and Technology, 60(2), 273–281. https://doi.org/10.2166/wst.2009.298.

Henonin, J., Russo, B., Mark, O., & Gourbesville, P. (2013). Real-time urban flood forecasting and modelling – a state of the art. Journal of Hydroinformatics, 15(3), 717–736. https://doi.org/10.2166/hydro.2013.132.

Newton, C., Jarman, D., Memon, F.A., Andoh, R., Butler, D. (2013): Implementation and assessment of critical input hyetograph generation methodology for use in a decision support tool for the design of flood attenuation systems. Proceedings from the International Conference on Flood Resilience, Exeter, 5-7 September 2013, pp 229-230.

---

## Author Response (AR1)

**Point-by-point response to the reviews**

**Anonymous Referee #1**

The authors conducted an interesting modeling experiment and presented a useful conclusion that updating drainage systems in response to urbanization is more effecient that updating in response to climate change.

I recommend that the authors address the following technical comments:

1. Page 12: when describing the changes in imperviousness, it should be expressed as x% points, not just x%. They talk about the changes in values expressed as percent, so x% points are appropriate.

Author response:
This has been changed accordingly.

2. Page 12 line 25: "for some areas, few models project a decrease in the intensity of extreme precipitation." –> you could make it more specific.

Author response:
This has been clarified following the reviewer's recommendation.

3. Page 15 line 12-13:" The impacts of uncertainties....(i.e. urban development)" –> I had a hard time figuring out.

Author response:
We thank the reviewer for pointing out that this sentence is a bit unclear. What is meant here is that the impact of uncertainties in urban development on flooding, that is, changes in impervious surfaces, is quantified by simulating the occurrence of the precipitation events using different assumptions about soil infiltration rates. In practice, we are using three different estimates of soil infiltration rates (average/low/high) for the individual cities as a means to visualize the potential uncertainty of soil sealing (i.e. in the context of this paper urban development is defined as changes in impervious surfaces). The text has been modified to make this clearer.

**Referee # 2 - S. Thorndahl (Referee)**

1. The authors present a novel approach to compare the impacts on urbanization and climate change on pluvial flooding in four European cities. The paper is well written and easily read and understood. Results are interesting in that sense that the impact of urbanization and climate change on flooding is in the same order of magnitude. There are some assumptions, which in my opinion need to be clarified in order to justify the simplifications of models and inputs. However since focusing on relative comparisons between the four cities and different climate scenarios, the effect of these assumptions on the final results and conclusions might be small - however still relevant to discuss from a scientific point of view.

Author response:

The authors agrees with the reviewer that some of the assumptions made in this paper would have been too simplistic for other applications, including for detailed planning of drainage system updates or for recommendations for location-specific adaptation measures. However, as the reviewer also notices, since the focus of the paper is on the relative comparison of the impacts of urban development and climate change on flood exposure between four European cities, the effect of these assumptions are not considered to affect the results, nor the conclusions, considerably. The authors agree with the reviewer that the importance and influence of some of the assumptions are highly relevant to discuss in a broader scientific context. In the following we will initiate this discussion, and, where appropriate, amend the manuscript accordingly. However, in order to preserve the focus on the paper we will focus on highlighting what assumptions are reasonable in the context of the study and avoid making (too many) diversions on other possible framings.

2.  In the abstract and the introduction the presence and impacts of an urban drainage system in terms of pluvial flooding is not mentioned. The first pages would benefit from a clarification on this.

Author response:

The authors agree with the reviewer that the paper will improve by including this information in the initial sections of the paper. A clarification on the presence and impacts of an urban drainage system in terms of pluvial flooding has been included in both the abstract and the introduction.

3.  In section 2.1 on the framework, it would be worth noting which type of precipitation input is used in the modelling concept.

Author response:

This information has been included in section 2.1.

4.  Line 1, page 6: Why the near-linear relationship. In eq.1 the relationship is linear.

Author response:

This is an error. The text will be changed from "near-linear" to "linear".

5.  I think the assumption on assuming fully saturated soils at all times, thus simplifying Horton, might need some more clarification. If you have a dry soil the initial infiltration capacity might easily be a decade larger than the end infiltration. In that case you risk overestimating the surface runoff. For some of the sandy soils, e.g. the example for Odense, has a saturated infiltration capacity of ~30 mm/h. If the initial capacity is ten times larges, it is doubtful that the soil will ever get saturated, since you would need rainfall intensities larger than 300 mm/h (for a longer period). But as you write in the discussion the problem decrease for larger return periods, thus larger rainfall intensities.

Author response:

The authors agree that this assumption may lead to overestimations of surface run-off and consequently exaggerated flood extents. Indeed, based on the work we present here we have made an analysis of typical soil and precipitation characteristics for Denmark and have found that in general the initial conditions will lead to changes in the runoff of approximately 1-6 mm depending on soil type and independent of return period (results are not published yet). However, since the assumption is consistent across all sites, soil

types, and return periods, and because we are primarily interested in comparing the influence of urban development and climate change on changes in flooding between the different urban areas, this assumption is not expected to influence the results. Instead, if the purpose of our analyses were to accurately estimate the precise area experiencing flooding during specific precipitation, a more detailed representation of infiltration processing, e.g. including initial losses should be applied. In addition, it should be noted that because infiltration varies considerably across small distances, infiltration data is associated with large uncertainties, and accurate location-specific infiltration data is thus not readily available for many locations. In our analyses, we attempt to address this uncertainly by simulating the precipitation events using low, average and high estimates of infiltration. We discuss this on page 8, lines 7-18. Given that we have not made the analysis of the importance of this assumption we have now added the word 'slightly' on line 14.

6. In the "method section", page 8, the assumption of subtracting the rainfall intensities with a 5 yr return period needs more clarification. It is discussed later on, but it could be relevant to discuss here also. I think neglecting the drainage system, by subtracting the rainfall from a 5 yr return period might be too simple an approach. Partly due to the fact that not all parts of the drainage systems might be designed for a 5 yr return period, and partly due to the fact that even designed for a 5 yr return period, there might be lots of local capacity left in the drainage system, depending on the rainfall dynamics. Furthermore, what about capacity of bassins, channels, recieving waters, etc.?

Author response:

We agree that a more complex model would yield more precise results if we collected the data needed to build and calibrate a full 1D2D model of both subsurface infrastructure, topography, and natural waterways. As mentioned in the beginning of this rebuttal, that might have been a reasonable approach if the objective of the study was to make detailed designs of adaptation measures. However, for the purpose of the study the methodology is perfectly valid and has been used before, e.g. (Arnbjerg-Nielsen & Fleischer, 2009) and is recommended as one of several simulation methods for both planning and real-time applications by (Henonin, Russo, Mark, & Gourbesville, 2013). The last, but also important, reason is that we do not have access to the data needed in order to make a 1D2D simulation more accurate than the method we have employed.

7. Page 12, top. The construction of precipitation events needs to be detailed. Are you using design storms e.g. the Chicago design storm constructed from the IDF-curves? These types of storms require a very linear rainfall- flood response in order to be valid. Has this been investigated?

Author response:

We do employ CDS-storms and have clarified this in the revised manuscript. We need this assumption first of all because we only have IDF-curves available at some of the sites we study. Our results are not a linear function of neither return period nor rainfall volume so we do not understand this part of the reviewers comment. The key assumption for CDS-storms is that a well-defined time of concentration exists, which we need to assume and think that this is reasonable in the current situation. Testing of this particular hypothesis was the focus of a presentation at the International Conference on Flood Resilience in Exeter (Newton et al, 2013). Because the proceedings only contain extended abstracts we have been in contact

with the authors regarding more detailed reporting of the results. Unfortunately, they have not published these results. The following text has been added to page 12, line 16: "… Chicago design storm…"

8. Also, why limit the duration to 4 hours – you might underestimate the role of storage bassins, and natural waterways, only looking at the very short durations, thus high intensity, rainfall.

Author response:

See also response to point no 6. We have highlighted that we only focus on pluvial flooding, i.e. assumed that the response is so short that flooding from larger waterways can be neglected.

9. Page 12. You state that you include the total rainfall amounts in the supplementary material, but I think it would be relevant to present here along with the max. intensities over different durations. In that case it would be possible to compare the infiltration rates to rainfall intensities.

Author response:

We agree with the point of not only presenting the infiltration rates and climate factors but also the actual rainfall intensities. We will therefore add a new figure to the manuscript (figure 4) showing maximum 1 hour intensity as well as total volume of the CDS storms for all return periods and all sites.

10. Section 3.4. I guess that you assume all surfaces to be impervious during the flood, meaning that you only account for the infiltrating water to the soil in the rainfall input to the model. In reality you might have a flood where water flows from impervious surface to pervious surfaces and infiltrated, but a I guess this is not accounted for. Please comment on this.

Author response:

Infiltration continues during the flood, meaning that water flowing from fully saturated surfaces to areas with excess capacity will infiltrate accordingly. To accommodate this, a net infiltration rate is defined in MIKE 21 (infiltration is accounted for as a separate input to the model, i.e. as an evaporation type2 file in MIKE 21 and not as part of the rainfall input) allowing for both precipitation and runoff to infiltrate at any stage during the events. This has now been clarified in the revised version of the manuscript.

11. Line 12, page 15: I think the limit of 10 cm water levels considered flooding needs some clarification. If you have grid cells of 25 x 25 m2 and with a min water level of 10 cm you discard 62.5 m3 of water. This is a significant amount! Please comment.

Author response:

We have studied several thresholds ranging from 1-50cm (not included in the paper) and found that our results are consistent independent of the selected threshold. Again, see response to point 6: our main focus is not on detailed calculation of the spatial distribution of the floods, but a comparative analysis of the drivers causing changes in flood hazards at city level.

12. In fig. 6 you use km2 and in fig. 7 you use ha. Please apply same units.

Author response:

Using the same units leads to very large or very small numbers in one of the graphs. We have therefore added the following text to the caption: Please note the differences in unit on the y-axis compared to Figure X to each of the figures.

Literature referred to in the author responses

Arnbjerg-Nielsen, K., & Fleischer, H. (2009). Feasible adaptation strategies for increased risk of flooding in cities due to climate change. *Water Science and Technology*, *60*(2), 273–281. https://doi.org/10.2166/wst.2009.298

Henonin, J., Russo, B., Mark, O., & Gourbesville, P. (2013). Real-time urban flood forecasting and modelling – a state of the art. *Journal of Hydroinformatics*, *15*(3), 717–736. https://doi.org/10.2166/hydro.2013.132

Newton, C., Jarman, D., Memon, F.A., Andoh, R., Butler, D. (2013): Implementation and assessment of critical input hyetograph generation methodology for use in a decision support tool for the design of flood attenuation systems. Proceedings from the International Conference on Flood Resilience, Exeter, 5-7 September 2013, pp 229-230.

**List of all relevant changes made in the manuscript**

**Abstract: page 1, line 24. The following text has been added:**

In addition, two different assumptions are examined with regards to the development of the capacity of the urban drainage system in response to urban development and climate change. In the "stationary" approach, the capacity resembles present-day design, while it is updated in the "evolutionary" approach to correspond to changes in imperviousness and precipitation intensities due to urban development and climate change respectively.

**Abstract: page 2, line 4. The following text has been added:**

Developing the capacity of the urban drainage system in relation to urban development is found to be an effective adaptation measure as it fully compensates for the increase in run-off caused by additional sealed surfaces. On the other hand, updating the drainage system according to changes in precipitations intensities caused by climate change only marginally reduce flooding for the most extreme events.

**Introduction: page 3, line 32. The following text has been added:**

We investigate the effectiveness of updating the urban drainage system in response to urban development and climate change simulating two scenarios; (1) the capacity of the drainage system is updated to correspond to changes in impervious surfaces and precipitation intensities, and (2) no modifications in the capacity of the drainage systems are assumed.

**Section 2.1: page 4, line 29. The following text has been added:**

High-intensity precipitation events with intensities corresponding to RPs of 10, 20, 50 and 100 years are included in the analysis.

**Section 2.2: page 6, line 9. The following text has been modified:**

Text has been changed from "*a near linear*" to "*a linear*".

**Section 2.2: page 6, line 11. The following text has been added:**

(Table S1 in Supplementary Materials)

**Section 2.2: page 6, line 20. The following text has been added:**

(Table S2 in Supplementary Materials)

**Section 2.3: page 8, line 14. The following word has been added:**

Slightly

**Section 2.3: page 9, line 7. The following text has been added:**

The runoff from each grid cell is allowed to infiltrate later during the events, i.e. runoff flowing from fully saturated surfaces to areas with excess capacity will infiltrate accordingly. To accommodate this, a net infiltration rate is defined in MIKE 21 allowing for infiltration of both precipitation and runoff at any stage during the events.

**Section 2.3: page 9, line 10. The following text has been modified:**

Text has been changed from *"is"* to *"can therefore be"*

**Section 2.5: page 12, line 16. The following text has been modified:**

Text has been changed from

*"IDF curves are used to construct time series of design precipitation events for 5, 10, 20, 50 and 100 RPs"*

to

*"IDF curves are used to construct time series of Chicago design storms  for 5, 10, 20, 50 and 100 year RPs"*

**Section 2.5: page 12, line 20. The following text has been modified:**

Text has been changed from

*"Figure S1 in the supplementary materials shows total precipitation for the different precipitation events and cities under present-day and future climatic conditions."*

to

*"Figure 4 shows maximum 1 hr intensity and total precipitation for the different precipitation events and cities under present-day and future climatic conditions."*

**Section 2.5: page 13, line 1. The following figure 4 has been inserted:**

[Figure]

**Figure 1: (a) total precipitation and (b) maximum 1 hr precipitation intensities during the individual design events for Nice, Strasbourg, Vienna and Odense. Total precipitation duration is four hours. Error bars for the RCP 4.5 and RCP 8.5 represent low/high climate factors (CFs) respectively (low CF = 10th percentile, high CF = 90th percentile).**

**Section 3.1: page 13, line 10. The following text has been modified:**

Text has been changed from

*"with absolute changes ranging from 6.6 % in Nice to 11.6 % in Strasbourg (**Error! Reference source not found.**)."*

to

*"….with absolute changes ranging from 6.6 % points in Nice to 11.6 % points in Strasbourg (**Error! Reference source not found.**)."*

**Section 3.2: page 14, line 13. The following text has been modified:**

Text has been changed from

*"For some areas, few models project a decrease in the intensity of extreme precipitation (CFs < 1)."*

to

*"For Vienna and Odense, few models project a decrease in the intensity of extreme precipitation (CFs < 1) for the RCP 4.5 scenario."*

**Section 3.2: Table 2. The following text has been modified:**

All instances of RCP45 in table 2 has been changed to RCP4.5

All instances of RCP85 in table 2 has been changed to RCP8.5

**Section 3.4: page 17, line 6. The following text has been modified:**

Text has been changed from

*"Uncertainties in the estimated impacts of urban development and climate change are examined by varying, respectively, potential soil infiltration rates and climate factors for each of the four cities and are 15 represented as error bars in all the following figures. The impacts of uncertainties related to the projection of future precipitation intensities (i.e. climate change) are comparable to those of accurately estimating soil-infiltration properties (i.e. urban development)."*

to

*"In all of the following figures, we show explicitly the results of changing the impervious surfaces and the simulated effect of climate change, and hence error bars representing uncertainties in the estimated impacts of urban development and climate change, respectively, are calculated based on variations in the potential soil infiltration rates and climate factors for each of the four cities. The range of uncertainties related to the projection of future precipitation intensities  are found to be largely comparable to those of soil infiltration estimates."*

**Section 3.4: page 20, line 3. The following text has been added to figure 8 caption:**

Please note the differences in unit (ha) on the y-axis compared to Figure 7 (km2).

**Supplementary materials**

Figure S1 has been removed from the supplementary materials, as it is now included in the manuscript (Figure 4).

[revised manuscript text omitted]